

# Cross-species screening of microsatellite markers for individual identification of snow petrel *Pagodroma nivea* and Wilson's storm petrel *Oceanites oceanicus* in Antarctica

Anant Pande[1],[*], Nidhi Rawat[2],[*], Kuppusamy Sivakumar[1],
Sambandam Sathyakumar[1], Vinod B. Mathur[3] and Samrat Mondol[2]

[1] Endangered Species Management, Wildlife Institute of India, Dehradun, Uttarakhand, India
[2] Animal Ecology and Conservation Biology, Wildlife Institute of India, Dehradun, Uttarakhand, India
[3] Wildlife Institute of India, Dehradun, Uttarakhand, India
[*] These authors contributed equally to this work.

Corresponding author
Samrat Mondol, samrat@wii.gov.in

## ABSTRACT

Seabirds are important indicators of marine ecosystem health. Species within the order Procellariiformes are the most abundant seabird species group distributed from warm tropical to cold temperate regions including Antarctica. There is a paucity of information on basic biology of the pelagic seabird species nesting on the Antarctic continents, and long-term studies are required to gather data on their population demography, genetics and other ecological parameters. Under the 'Biology and Environmental Sciences' component of the Indian Antarctic programme, long-term monitoring of Antarctic biodiversity is being conducted. In this paper, we describe results of cross-species screening of a panel of 12 and 10 microsatellite markers in two relatively little studied seabird species in Antarctica, the snow petrel *Pagodroma nivea* and the Wilson's storm petrel *Oceanites oceanicus*, respectively. These loci showed high amplification success and moderate levels of polymorphism in snow petrel (mean no. of alleles 7.08 ± 3.01 and mean observed heterozygosity 0.35 ± 0.23), but low polymorphism in Wilson's storm petrel (mean no. of alleles 3.9 ± 1.3 and mean observed heterozygosity 0.28 ± 0.18). The results demonstrate that these panels can unambiguously identify individuals of both species (cumulative $PID_{sibs}$ for snow petrel is $3.7 \times 10^{-03}$ and Wilson's storm petrel is $1.9 \times 10^{-02}$) from field-collected samples. This work forms a baseline for undertaking long-term genetic research of these Antarctic seabird species and provides critical insights into their population genetics.

## INTRODUCTION

As top predators, seabirds maintain the structure of marine food webs, regulate island and marine ecosystem processes and act as indicators of marine ecosystem health (*Lascelles et al., 2012*; *Paleczny et al., 2015*). Their ability to fly over large distances,

their extreme life history strategies (monogamy, slow reproduction, late sexual maturity), natal philopatry, high visibility and dependence on land for breeding make it essential to conduct long-term population level studies (*Piatt, Sydeman & Wiese, 2007*) for better understanding of their biology. Recent studies focusing on seabird population monitoring have highlighted the threatened status of seabirds across the globe (*Croxall et al., 2012*), especially in the Southern Ocean where seabird populations have declined substantially over last few decades (*Paleczny et al., 2015*). This has led to interdisciplinary approaches to understand seabird population dynamics in order to aid their conservation and management across their distribution range (*Croxall et al., 2012*; *Taylor & Friesen, 2012*).

Seabirds within the order Procellariiformes comprising petrels, shearwaters, albatrosses, storm petrels, and diving petrels represent one of the most widely distributed and abundant avifauna (*Warham, 1996*). Despite their broad distribution and large population sizes, long-term ecological and genetic data exists for few of these species across the globe. In addition to several ecological studies on Procellariiformes (*Croxall et al., 2012*), some recent studies have used genetic data to address important biological parameters such as relatedness, population structure, past population demography (e.g. see *Gómez-Díaz, González-Solís & Peinado, 2009* for Cory's shearwater; *Welch et al., 2012* for Hawaiian petrel) for species distributed in tropical and Arctic marine ecosystems. Research on the biology of Procellariiformes is relatively limited in the Southern Ocean ecosystem, especially in Antarctica because of its remoteness and associated logistical difficulties. Despite site-specific monitoring of some Procellariiformes on sub-Antarctic islands (e.g. *Brown et al., 2015* for giant petrels; *Quillfeldt et al., 2017* for Antarctic prion, thin-billed prion and blue petrel) and the Antarctic coast (e.g. *Barbraud & Weimerskirch, 2001* for snow petrel; *Barbraud & Weimerskirch, 2006* for multiple species; *Techow et al., 2010* for giant petrels), long-term ecological as well as genetic research is sparse. *Nunn & Stanley (1998)* reported the phylogenetic relationships among Procellariiformes using a neighbour-joining approach, but within each family, detailed population genetic information is lacking. Prior studies have used Restriction Fragment Length Polymorphisms and allozymes to investigate genetic variation and extra-pair paternity in snow petrel as well as some other Procellariiformes (*Jouventin & Viot, 1985*; *Viot, Jouventin & Bried, 1993*; *Lorensten et al., 2000*, *Quillfeldt et al., 2001*) in Antarctica.

As part of the 'Biology and Environmental Sciences' component of the Indian Antarctic Program, we conducted comprehensive ecological surveys between 2009 and 2016 to understand seabird and marine mammal ecology around the Indian Antarctic research stations (*Pande et al., 2017*). Currently, this programme is focused on generating baseline genetic data of breeding seabird species found around Indian area of operations in Antarctica, especially on snow petrel *Pagodroma nivea* and Wilson's storm petrel *Oceanites oceanicus*. The snow petrel is endemic to Antarctica and Southern Ocean with breeding distribution along Antarctic coast including some inland mountains and few sub-Antarctic islands (*Croxall et al., 1995*). On the other hand, the Wilson's storm petrel has a much wider breeding distribution from Cape Horn to the Kerguelen Islands and coastal Antarctica and migrates to the mid-latitudes of the north Atlantic, north
Indian and Pacific Oceans during non-breeding period (*BirdLife International, 2017*). Effective monitoring of these species in the Indian Antarctic sector will require systematic information on their distribution, current population status and genetic parameters. In this paper, we report results from cross-species screening of microsatellite markers for individual identification of snow petrel and Wilson's storm petrel in Antarctica. These tested microsatellite panels will provide valuable tools for estimating levels of genetic variation, relatedness, and genetic signals of population demography, in both species across their ranges.

## METHODS

### Study area

We carried out sampling at Larsemann Hills, Prydz Bay and Schirmacher Oasis, Central Dronning Maud Land (Fig. 1); close to permanent Indian research stations in Antarctica *Bharati* (Larsemann Hills) and *Maitri* (Schirmacher Oasis). Distance between these two study areas is about 2,500 km. Larsemann Hills (69°20′S–69°30′S; 75°55′E–76°30′E), are a group of islands in Prydz Bay located on the Ingrid Christensen Coast, Princess Elizabeth Land of east Antarctica. This island group comprises of variously sized islands and peninsulas, located halfway between the eastern extremity of the Amery Ice Shelf and the southern boundary of the Vestfold Hills. Schirmacher Oasis, Central Dronning Maud Land (70°44′–70°46′S and 11°22′–11°54′E) is situated on the Princess Astrid coast about 120 km from the Fimbul ice shelf. Four species of seabirds (Adelie penguin *Pygoscelis adeliae*, southpolar skua *Stercorarius maccormickii*, snow petrel and Wilson's storm petrel) breed in the ice-free areas of Larsemann Hills, whereas only the south polar skua breeds at Schirmacher Oasis (*Pande et al., 2017*).

### Field sampling

We conducted sampling for this study as part of the 'Antarctic Wildlife Monitoring Programme' under the Indian Scientific Expedition to Antarctica (Expedition nos. 33, 34, and 35) during the austral summers (November–March) of 2013–14, 2014–15, and 2015–16. We adopted an opportunistic genetic sampling approach under the seabird nest monitoring protocol (see *Pande et al., 2017*) for snow petrel sample collection. First, we selected previously marked nest sites with breeding snow petrel individuals for genetic sampling. Subsequently, we conducted both non-destructive (buccal swabs and blood smears) and non-invasive (hatched eggshells and abandoned eggs) sampling to collect biological materials from the monitored nesting sites. During non-destructive sampling of snow petrel individuals, we carefully hand-captured birds at their nest cavities and collected buccal swabs or blood samples. We collected blood samples from bird's brachial vein using 0.1 ml sterilized syringe needles and stored in EDTA vacutainer tubes. We collected buccal epithelial tissue by gently rotating a sterilized cotton-tipped swab against the inner cheek of the bird (*Handel et al., 2006*). All individuals were released within 60 s of capture. Wherever available, we also collected hatched eggshells, shed adult feathers and abandoned eggs from the nests. In addition, we also conducted opportunistic muscle tissue sampling of snow petrel carcasses from wherever they were found. These

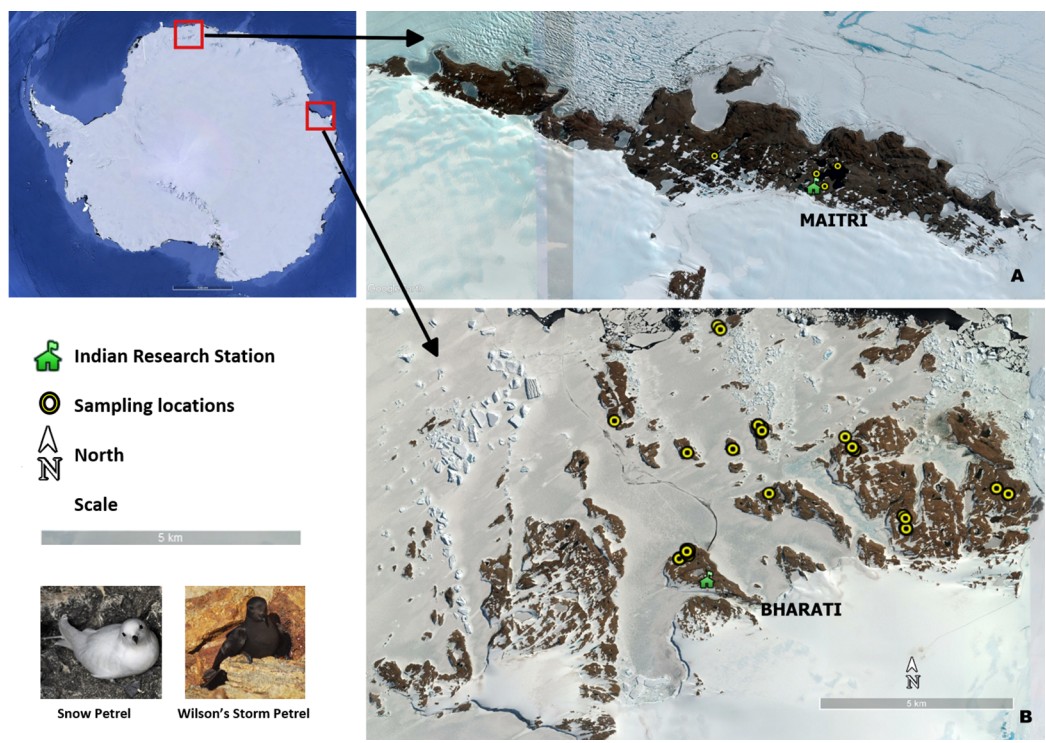

**Figure 1 Seabird sampling locations in Antarctica.** (A) Schirmacher oasis, site of *Maitri* station (B) Larsemann hills, site of *Bharati* station (Photo Credit: Anant Pande).

carcasses were mostly from birds predated by south polar skua or found naturally dead due to other unidentified causes. We did not find any nesting sites of snow petrels at Schirmacher Oasis during our field surveys, and thus opportunistic sampling of carcasses for muscle tissue was conducted.

Similarly, we collected Wilson's storm petrel samples from monitored nesting sites at Larsemann Hills. All genetic samples of Wilson's storm petrel were collected opportunistically through carcass muscle tissue collection as capturing them was not possible due to their narrow nest cavities. No Wilson's storm petrel samples were collected from Schirmacher Oasis. We stored the samples collected at field sites at −20 °C at respective Indian Antarctic research stations before being brought to Wildlife Institute of India, Dehradun for further laboratory analysis. The details of sampling locations are provided in File S1.

## Primer selection

As there are no species-specific microsatellite markers published for snow petrel, we screened a panel of cross-species markers for individual identification of snow petrels. We selected 15 microsatellite markers developed for Hawaiian petrel *Pterodroma sandwichensis* (Nine markers, *Welch & Fleischer, 2011*) and white-chinned petrel *Procellaria aequinoctialis* (Six markers, *Techow & O'Ryan, 2004*). These markers were selected based on their polymorphic information content (number of alleles as well as

expected heterozygosity) in the aforementioned species. We also tested this panel of 15 microsatellite loci for individual identification of Wilson's Storm Petrel samples.

## DNA extraction and primer standardization

We used muscle tissue samples of snow petrel and Wilson's storm petrel for initial standardization and validation of microsatellite panel. Genomic DNA was extracted in duplicate from all tissue samples using commercially available DNeasy Tissue kit (Qiagen Inc., Valencia, CA, USA) using a modified approach. In brief, all samples were macerated with sterile blades independently, followed by overnight complete tissue digestion with 25 μL proteinase-K. Post-digestion, extraction was performed using Qiagen animal tissue spin column protocol. DNA was eluted twice with 100 μL of $1 \times$ TE and stored in $-20\,^{\circ}C$ until further processing. Each set of 11 extractions was accompanied with one extraction control to monitor possible contamination.

We conducted all initial PCR standardizations using muscle tissue DNA samples. We carried out amplifications for each primer in 10 μL reaction volumes containing four μL Qiagen Multiplex PCR buffer mix (Qiagen Inc., Valencia, CA, USA), 0.2 μM labeled forward primer, 0.2 μM reverse primer, four μM BSA, and two μL of 1:10 diluted DNA extract. The temperature regime included an initial denaturation ($94\,^{\circ}C$ for 15 min); 35 cycles of denaturation ($94\,^{\circ}C$ for 30 s), annealing (53 or $57\,^{\circ}C$ for 45 s) and extension ($72\,^{\circ}C$ for 45 s); followed by a final extension ($72\,^{\circ}C$ for 30 min). Post-temperature standardization, primers with identical annealing temperatures was optimized for multiplex reactions with the same samples of both species (see Table 1). Subsequently, all test samples were amplified with standardized parameters. During all amplifications, both extraction controls and PCR negative controls (one PCR negative every set of amplifications) were included to monitor any possible contamination. PCR products were visualized in 2% agarose gels, and genotyped using LIZ500 size standard in an automated ABI3500XL genetic analyser. Microsatellite alleles were scored using program GENEMARKER (Softgenetics Inc., State College, PA, USA) and allele bins for each locus were created from the data generated. We randomly re-genotyped 15% of each locus from different samples to check for reliable genotypes and estimated genotyping error rates.

## Data analysis

We calculated average amplification success as the percent positive PCR for each locus, as described by *Broquet, Ménard & Petit (2007)*. We quantified allelic dropout and false allele rates manually as the number of dropouts or false alleles over the total number of amplifications, respectively (*Broquet, Ménard & Petit, 2007*). We also calculated the Probability of Identity for siblings ($PID_{sibs}$), the probability of two individuals drawn from a population sharing the same genotype at multiple loci and the theoretical Probability of Identity or $PID_{unbiased}$ (*Waits, Luikart & Taberlet, 2001*) using program GIMLET (*Valière, 2002*). We tested the frequency of null alleles in our data set using FREENA (*Chapuis & Estoup, 2007*) whereas summary statistics, tests for deviations from Hardy–Weinberg equilibrium and pairwise linkage disequilibrium were calculated for each locus using program ARLEQUIN v.3.1 (*Excoffier, Laval & Schneider, 2005*).

**Table 1 Characterization of microsatellite loci genotyped in snow petrel and Wilson's storm petrel individuals from Antarctica.**

| Sp. | Locus | Primer sequences 5'–3' | Repeat nature | Repeat motif | Dye | PSR (bp) | Ta °C | Na | Ho | He | Allelic range | PID$_{unbiased}$ cumulative | PID$_{sibs}$ cumulative | AS (%) | ADO (%) | PCR set |
|---|---|---|---|---|---|---|---|---|---|---|---|---|---|---|---|---|
| **a) Snow Petrel (n = 55)** | Ptero08[a] | F: GCACCTGCTGGTGATGAGTC R: AGGGAAAAGGAAACCATCCAG | Tetra | (AAGG)$_8$ | VIC | 181–221 | 53 | 11 | 0.49 | 0.73 | 52 | $8.03 \times 10^{-02}$ | $4.1 \times 10^{-01}$ | 96.4 | 0 | Set 2 |
| | Paequ03[b] | F: TGTGGGTGCAGTAGAGCA R: CAATAAGAAGATCAGCAGAACAGAC | Di | (GA)$_{19}$ | VIC | 219–243 | 53 | 12 | 0.68 | 0.72 | 24 | $7.63 \times 10^{-03}$ | $1.71 \times 10^{-01}$ | 98.2 | 0 | Set 1 |
| | Ptero07[a] | F: TTAAAAATCGGTCCAATAGTCG R: GCACAGAGTTGACCGTGTTG | Tetra | (AAAG)$_8$ | FAM | 177–217 | 53 | 8 | 0.53 | 0.66 | 48 | $1.12 \times 10^{-03}$ | $7.89 \times 10^{-02}$ | 98.2 | 3.6 | Set 1 |
| | Ptero04[a] | F: TGCATTGTTTCTGTCCAAACTC R: GGCTGGAATGCATAGTACCAAC | Di | (CA)$_{13}$ | FAM | 117–147 | 53 | 11 | 0.67 | 0.63 | 32 | $1.81 \times 10^{-04}$ | $3.78 \times 10^{-02}$ | 100 | 0 | Set 2 |
| | Ptero10[b] | F: GAAGCTGCACTGGAACTG R: CATGTTGGTAAGAATCCAGATG | Di | (CA)$_8$ | NED | 159–183 | 53 | 7 | 0.20 | 0.56 | 12 | $3.79 \times 10^{-05}$ | $1.99 \times 10^{-02}$ | 98.2 | 0 | Set 2 |
| | Paequ13[b] | F: GACCTGCAGCAATAGCACGAC R: TGCCTTCATCAGAATCCTCCTG | Di | (GT)$_9$ | PET | 144–150 | 57 | 4 | 0.07 | 0.44 | 6 | $1.27 \times 10^{-05}$ | $1.23 \times 10^{-02}$ | 100 | 0 | Set 3 |
| | Paequ07[b] | F: TGCAGACCTGACTTTCACAGCTC R: CCTCCAAACATCCAGCCATC | Di | (GT)$_{12}$ | FAM | 314–320 | 57 | 3 | 0.30 | 0.40 | 6 | $4.83 \times 10^{-06}$ | $7.92 \times 10^{-03}$ | 100 | 0 | Set 3 |
| | Paequ02[b] | F: GCCTACTCCATCTTAATTGTG R: GGTTCATACAGTTTCCTAGGTC | Di | (CA)$_2$TT(CA)$_{10}$ | PET | 180–200 | 53 | 7 | 0.03 | 0.30 | 30 | $2.32 \times 10^{-06}$ | $5.74 \times 10^{-03}$ | 98.2 | 1.8 | Set 2 |
| | Ptero03[b] | F: TGTGTACAGCATGTGCTTGAG R: GCTGAATGGCAGTTTCTTCC | Di | (CA)$_9$ | FAM | 165–177 | 53 | 4 | 0.10 | 0.23 | 22 | $1.36 \times 10^{-06}$ | $4.50 \times 10^{-03}$ | 100 | 0 | Set 1 |
| | Paequ08[b] | F: TATTCTGAGACTTGCGTTATCC R: GTGATCCATTAGTTGATGTCTACTG | Di | (CA)$_{11}$ | PET | 215–223 | 53 | 4 | 0.16 | 0.18 | 8 | $8.89 \times 10^{-07}$ | $3.71 \times 10^{-03}$ | 100 | 0 | Set 1 |
| | *Ptero09[a] | F: GCAAATACCAGTCTTCCAAAGG R: TTTAAGATAAAGATGTTTGAGAACCAC | Tetra | (AAGG)$_8$ | FAM | 161–189 | 57 | 9 | 0.67 | 0.72 | 28 | – | – | 100 | 0 | Set 3 |
| | *Ptero01[a] | F: GAAAACAACTCCCCACAAC R: TCCGTCAGACCTGCTGTATG | Di | (CA)$_7$ | PET | 82–104 | 53 | 5 | 0.33 | 0.32 | 24 | – | – | 98.2 | 0 | Set 1 |
| | **Mean(SD)** | | | | | | | 7.08 (3.01) | 0.35 (0.23) | 0.49 (0.19) | 24.5 (14.5) | | | | | |
| **b) Wilson's Storm Petrel (n = 24)** | Ptero07[a] | F: TTAAAAATCGGTCCAATAGTCG R: GCACAGAGTTGACCGTGTTG | Tetra | (AAAG)$_8$ | FAM | 177–217 | 53 | 6 | 0.42 | 0.76 | 40 | $7.67 \times 10^{-02}$ | $3.95 \times 10^{-01}$ | 100 | 0 | Set 1 |
| | Paequ10[b] | F: GAAGCTGCACTGGAACTG R: CATGTTGGTAAGAATCCAGATG | Di | (CA)$_8$ | NED | 181–191 | 53 | 4 | 0.38 | 0.64 | 10 | $1.39 \times 10^{-02}$ | $1.91 \times 10^{-01}$ | 100 | 0 | Set 1 |
| | Paequ13[b] | F: GACCTGCAGCAATAGCACGAC R: TGCCTTCATCAGAATCCTCCTG | Di | (GT)$_9$ | PET | 146–148 | 57 | 2 | 0.08 | 0.5 | 2 | $4.59 \times 10^{-03}$ | $1.13 \times 10^{-02}$ | 100 | 8.3 | Set 2 |
| | Paequ08[b] | F: TATTCTGAGACTTGCGTTATCC R: GTGATCCATTAGTTGATGTCTACTG | Di | (CA)$_{11}$ | PET | 219–227 | 51 | 3 | 0.21 | 0.47 | 8 | $1.58 \times 10^{-03}$ | $6.91 \times 10^{-02}$ | 100 | 0 | Set 3 |
| | Ptero01[a] | F: GAAAACAACTCCCCACAAC R: TCCGTCAGACCTGCTGTATG | Di | (CA)$_7$ | PET | 165–177 | 53 | 4 | 0.17 | 0.44 | 12 | $5.56 \times 10^{-04}$ | $4.28 \times 10^{-02}$ | 100 | 0 | Set 1 |
| | Paequ03[b] | F: TGTGGGTGCAGTAGAGCA R: CAATAAGAAGATCAGCAGAACAGAC | Di | (GA)$_{19}$ | VIC | 219–235 | 53 | 5 | 0.21 | 0.39 | 16 | $2.04 \times 10^{-04}$ | $2.79 \times 10^{-02}$ | 100 | 0 | Set 1 |
| | Ptero03[a] | F: TGTGTACAGCATGTGCTTGAG R: GCTGAATGGCAGTTTCTTCC | Di | (CA)$_9$ | FAM | 88–104 | 57 | 2 | 0.17 | 0.35 | 16 | $9.25 \times 10^{-05}$ | $1.94 \times 10^{-02}$ | 91.7 | 0 | Set 2 |
| | *Paequ07[b] | F: TGCAGACCTGACTTTCACAGCTC R: CCTCCAAACATCCAGCCATC | Di | (GT)$_{12}$ | FAM | 312–318 | 51 | 3 | 0.08 | 0.16 | 6 | – | – | 100 | 4.2 | Set 3 |
| | *Ptero09[a] | F: GCAAATACCAGTCTTCCAAAGG R: TTTAAGATAAAGATGTTTGAGAACCAC | Tetra | (AAGG)$_8$ | FAM | 173–185 | 61 | 6 | 0.67 | 0.55 | 16 | – | – | 91.7 | 0 | Set 3 |
| | *Ptero04[a] | F: TGCATTGTTTCTGTCCAAACTC R: GGCTGGAATGCATAGTACCAAC | Di | (CA)$_{13}$ | FAM | 127–139 | 57 | 4 | 0.38 | 0.52 | 12 | – | – | 100 | 0 | Set 2 |
| | **Mean(SD)** | | | | | | | 3.9 (1.3) | 0.28 (0.18) | 0.48 (0.15) | 13.8 (9.7) | | | | | |

**Notes:**

Sp, species; PSR, product size range; Ta, annealing temperature; Na, number of alleles; Ho, observed heterozygosity; He, expected heterozygosity; PID, probability of identification; AS, amplification success; ADO, allelic dropout.

\* Locus deviating from Hardy–Weinberg equilibrium.
[a] *Welch & Fleischer (2011)*.
[b] *Techow & O'Ryan (2004)*.

## RESULTS AND DISCUSSION

We genotyped a total of 55 snow petrel and 24 Wilson's storm petrel samples to test and standardize the selected microsatellite markers. Snow petrel samples were selected from blood ($n = 1$), buccal swab ($n = 2$), carcass muscle tissue ($n = 24$), and hatched eggshells ($n = 28$) to test amplification success from different types of biological samples. Wilson's storm petrel samples were all from muscle tissue of individual carcasses collected in the field.

Of the 15 loci tested during the initial standardization, 12 loci showed amplification for snow petrel (loci Ptero2, Ptero6, and Ptero10 did not amplify), whereas only 10 loci successfully amplified for Wilson's storm petrel (loci Paequ2, Ptero2, Ptero6, Ptero8, and Ptero10 did not amplify) (Table 1). Subsequently, these panels of 12 and 10 loci were tested with all snow petrel and Wilson's storm petrel samples, respectively. Overall, the amplification success ranged between 96.4% and 100% for snow petrel and 91.7%–100% for Wilson's storm petrel; and allelic dropout rates were 0–3.6% and 0–8.3% for snow petrel and Wilson's storm petrel respectively (see Table 1 for more details). For snow petrel, the loci varied from highly polymorphic (Paequ03-12 alleles, $H_o$-0.68) to less polymorphic (Paequ13-4 alleles, $H_o$-0.07), whereas for Wilson's storm petrel the loci were moderately polymorphic (Ptero07-6 alleles, $H_o$-0.76) to less polymorphic (Paequ13-2 alleles, $H_o$-0.08) (Table 1). Two loci in snow petrel (Ptero01 and Ptero09) and three loci in Wilson's storm petrel (Ptero04, Ptero09, and Paequ07) were found to deviate from the Hardy–Weinberg Equilibrium, and thus were removed from any further analysis. There was no evidence for a strong linkage disequilibrium between any pair of loci (details in Files S2 and S3). $PID_{sibs}$ and $PID_{unbiased}$ values were found to be $3.71 \times 10^{-03}$ and $8.89 \times 10^{-07}$ for snow petrel, and $1.94 \times 10^{-02}$ and $9.25 \times 10^{-05}$ for Wilson's storm petrel, respectively. Locus-wise and average values for observed and expected heterozygosity, number of alleles and allelic size ranges are presented in Table 1. The frequency of null alleles across the loci was observed to be low in both the study species (snow petrel—$0.11 \pm 0.09$ and Wilson's storm petrel—$0.15 \pm 0.07$, respectively; see File S4).

This paper is the first attempt to use nuclear microsatellite markers to individually identify both snow petrel and Wilson's storm petrel in Antarctica, and the final panel described here provide unambiguous individual identification from both species in our study area. Testing the markers on various types of biological materials (tissue, blood, buccal swab, and hatched eggshells) showed high amplification success, but further tests with non-invasive samples (for example shed feathers) is required for long-term studies. Both $PID_{unbiased}$ and $PID_{sibs}$ values are also sufficient enough for population genetic studies considering low population sizes of snow petrel (800–1,000 individuals) and Wilson's storm petrel (<200 individuals) in the study area (*Pande et al., 2017*, A. Pande et al., 2018, unpublished data). However, it is important to point out that in case of Wilson's storm petrel the $PID_{sibs}$ value of $1.9 \times 10^{-2}$ may not be sufficiently robust enough for large populations based on suggestions of earlier studies (*Waits, Luikart & Taberlet, 2001*; *Miller, Joyce & Waits, 2005*), where such low value could potentially lead to higher misidentification rates. Future studies should consider testing additional cross-specific markers previously developed for prion species (*Pachyptila* sp.) but also tested in Wilson's

storm petrel (*Moodley et al., 2015*) to create a comprehensive panel for studying large populations of the species.

## CONCLUSION

Molecular genetic analysis has become crucial in understanding levels of genetic differentiation, hybridisation and extinction risk in seabird populations (*Taylor & Friesen, 2012*). In critical ecosystems such as Antarctica, individual-level genetic data can be a valuable tool to study evolution, adaptation, past events of diversifications and extinctions for wide-ranging seabirds. Moreover, genetic datasets on species of pelagic seabirds such as snow petrel and Wilson's storm petrel are generally lacking in comparison with charismatic species such as penguins. In this study, we could establish the efficacy of cross-species markers in individual identification of these two common Antarctic seabird species. In the future, we aim to build upon the long-term genetic research under the 'Antarctica Wildlife Monitoring Programme' by increasing spatio-temporal sampling efforts to understand the population structure, relatedness and other aspects and provide insights to seabird behaviour (monogamy, extra-pair paternity etc.) and evolution. This detailed genetic research will also aid in long-term ecological monitoring and conservation management of breeding seabird populations of Antarctica.

### Permits and ethical clearances

All samples were collected under the 'Biology and Environmental Sciences' component (Letter no: NCAOR/ANT/ASPA/2014-15/01) of the Indian Scientific Expeditions to Antarctica with appropriate approvals from the Environment Officer, Committee for Environmental Protection (Antarctic Treaty Secretariat), National Centre for Antarctic and Ocean Research, Earth System Science Organisation, Ministry of Earth Sciences, Government of India, Goa, India.

## ACKNOWLEDGEMENTS

We thank the National Centre for Antarctic and Ocean Research, Ministry of Earth Sciences for providing all logistic support during the Indian Scientific Expeditions to Antarctica. We are grateful to respective expedition leaders and team member volunteers of 33rd, 34th, and 35th Indian Scientific Expeditions to Antarctica for their assistance during field work. We thank A. Madhanraj, and MEERCAT lab members for their help in laboratory and Srinivas for his inputs on data analysis. We sincerely thank Wildlife Forensics and Conservation Genetics Cell, CAMPA Cell, Research Coordinator and Dean, Wildlife Institute of India for their support. Our sincere thanks to all the reviewers of the earlier version of the manuscript.

### Funding

The Wildlife Institute of India and DST-Inspire Faculty Award to Samrat Mondol (Grant no: IFA12-LSBM-47) provided financial support for this study. The funders had

no role in study design, data collection and analysis, decision to publish, or preparation of the manuscript.

## Grant Disclosures

The following grant information was disclosed by the authors:
Wildlife Institute of India and DST-Inspire Faculty Award to Samrat Mondol: IFA12-LSBM-47.

## Competing Interests

The authors declare that they have no competing interests.

## Author Contributions

- Anant Pande conceived and designed the experiments, performed the experiments, analysed the data, prepared figures and/or tables, authored or reviewed drafts of the paper, approved the final draft.
- Nidhi Rawat performed the experiments, analysed the data, authored or reviewed drafts of the paper.
- Kuppusamy Sivakumar conceived and designed the experiments, contributed reagents/materials/analysis tools, authored or reviewed drafts of the paper, approved the final draft.
- Sambandam Sathyakumar contributed reagents/materials/analysis tools, authored or reviewed drafts of the paper.
- Vinod B. Mathur contributed reagents/materials/analysis tools, authored or reviewed drafts of the paper, approved the final draft.
- Samrat Mondol conceived and designed the experiments, performed the experiments, analysed the data, contributed reagents/materials/analysis tools, prepared figures and/or tables, authored or reviewed drafts of the paper, approved the final draft.

## Animal Ethics

The following information was supplied relating to ethical approvals (i.e., approving body and any reference numbers):

All samples were collected under the 'Biology and Environmental Sciences' component (Letter no: NCAOR/ANT/ASPA/2014-15/01) of the Indian Scientific Expeditions to Antarctica with appropriate approvals from the Environment Officer, Committee for Environmental Protection (Antarctic Treaty Secretariat), National Centre for Antarctic and Ocean Research, Earth System Science Organization, Ministry of Earth Sciences, Government of India, Goa, India.

## Data Availability

Dryad: DOI:10.5061/dryad.57027hc.

## Supplemental Information

Supplemental information for this article can be found online at http://dx.doi.org/10.7717/peerj.5243#supplemental-information.

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
