# Peer review of "Cross-species screening of microsatellite markers for individual identification of snow petrel Pagodroma nivea and Wilson's storm petrel Oceanites oceanicus in Antarctica"

_PeerJ, doi:10.7717/peerj.5243_

## Round 0.1 · original submission · Major Revisions

A comment on the sample sizes sufficiency and limitations (in one taxon n=24, less than a half of the other one) is requested.

It is not clear to me which primers were used and their sequence or reference(s); Table 1 refers Sets 1-3 which I could not find in the text.
With very few marginal exceptions, most loci show (in many cases surely significant) observed heterozygosity values much lower than expected, suggesting frequent silent alleles.

There is an excessive proportion of dinucleotide STRs, which is not recommended given their typing problems.

Table 1 requires legend; allelic range (nº repeats?); repeat motif?
Last but not least, in order to be able to properly assess the validity of conclusions, pheno/genotype distributions (obs and expected) per locus and HW tests as well as LD/association tests results are required and should be added as supplemental material. A couple of figures depicting typical electropherograms would also be most welcome.

Reviewer 1 ·

Basic reporting

Professional English is used, but additional editing would be helpful to improve grammar, mostly in the Introduction.

Otherwise, basic reporting is acceptable. I have made some comments on certain portions of the writing throughout, which are included with this review.

Experimental design

The study establishes baseline data for population genetic studies of two Antarctic seabird species. These data are necessary foundations upon which to build more in-depth studies.

Credentials and permits appear adequate.

All methods are well-described, including deviations from published Qiagen protocols, though clarification is necessary on whether siblings were sampled in the case of the snow petrels, and if so, whether this was accounted for in analysis.

Validity of the findings

All results appear sound, though attention should be given to the point I raise in the previous section.

Additional comments

I am including comments that I feel would improve the overall quality of the writing. Thank you for the opportunity to contribute to this manuscript!

Annotated reviews are not available for download in order to protect the identity of reviewers who chose to remain anonymous.

·

Basic reporting

The manuscript entitled “Standardizing microsatellite panels for individual identification of seabirds’ Snow Petrel Pagodroma nivea and Wilson’s Storm Petrel Oceanites oceanicus in Antarctic” of Pande et al. have relevant information to monitoring Petrels form long-term in Antarctic region. However, the text need to be improved in many parts to became clearer and fluid. I do some suggestion, but I recommend to revised by native speaker English.
I did some suggestions about the structure of topics in the text, including delete some parts and change positions odf anothers.
Some parts the authors affirmed that are few studies with Procellariformes (lines 68-69, lines 73-80), being this study a pioneer with Procellariformes in Antarctica, but there are many genetics studies in Southern ocean. Thus, i recommended that authors change the focus this sentence, thus cited theses studies in Antartctica and draw attention the scarce studies with Snow Petrel and Wilson Storm Petrel.
Information about Indian Antarctic programme do not should be included in the text, can be included in the end together of permits of the study.

Experimental design

In general, the methods and results were appropriated and well conducted. The objective of the study included assessing population genetic structure. However, the present study aim developed a panel of microsatellites markers, it did not population structure analysis. Thus, i recommend take out population structure of objectives.

Validity of the findings

The results are valid and important, seem robust and conclusives

Additional comments

Major comments:
1- Information about Indian Antarctic programme do not should be included in the text, can be included in the end together of permits of the study.
3- Common names have start initial letter in capital letter Snow Petrel and Wilson’s Storm Petrel, thus fix the text.
Line 1- In the title delete “Standardizing “
The abstract need to be rewritten to be clearer.
Lines 30 -32 I suggest change the phrase . “Procellariiformes are one of the most abundant seabird, occurring from tropical to polar regions.”
Lines 32- 35 Change the sentence “Long-term studies with seabirds in Antarctica Continent is scarce, thus genetic tools to allow recovery population demography, evolutionary history of the species.”
Lines 35-36 I suggest deleted phrased “Under the ‘Biology and Environmental Sciences’ component of the Indian Antarctic programme, long-term monitoring of Antarctic biodiversity is being conducted.” It is irrelevant information and is contradictory with anterior phrase.
Lines 39-40 What is the percentage of amplification success?
Key-word – delete Indian Antarctic programme and Antarctic biodiversity; and include genetic diversity
Lines 58- change “makes” to “make”
Lines 87-89. This part of the text need to be altered, because it is not a reported is a manuscript. Thus, I suggested altered to “understanding distribution, abundance, population dynamics and genetics of Antarctic seabirds it is important to biodiversity conservation of the continent. “
Lines 89-94 – Delete this sentence, is not part of the present study. It is a aim of Indian Programme.
Lines 102- Change “With the broad objective of assessing population genetic structure. We describe a panel of cross-species microsatellite markers” to “The present study aim developed a panel of microsatellites markers…..” . You did not population structure analysis.
Lines 120 and 124 – delete coordinates word.
Lines 133- 135 – deleted “ as part of the ‘Antarctic Wildlife Monitoring Programme’ under the Indian Scientific Expedition to Antarctica (Expedition nos. 33, 34 and 35).” This information yet is found in Aknowdelege
Line 139- Change non-destructive to invase samples
Lines 167- 170 – It would be part of discussion.
Line 157_ What is the size of samples (number of individuals sampled) to each of samples and to species? The information on line 207 to 211 could be to here.
Lines 172-179 It would be after Field Samples. I suggested to one section of Field Sampled, DNA extraction, Primer selection and standardization
Line 215_ delete “see” and “for details”
Line 220- delete “see” and “for detailed summary statistics”
Change the order, lines 222-224 ( Overall….. until ….. Wilson’s storm petrel respectively.) before lines 216-222(For Snow Petrel …….. until ……pair of loci)
Lines 247-248 What the author want said with “petrel and Wilson’s storm petrel are generally lacking in comparison with charismatic species such as penguins”? It is not clearer, it is related with number of studies or diversity index?
Line 249 – 253. The future aims is not part of present study, thus I recommend change the text “ Markers developed in this study will be extremely important to future Petrels monitoring to understand the population structure, relatedness, behavior (monogamy, extra-pair paternity etc.) and evolution.”
Lines 108 -114- Permits and ethical clearances, transfer to the end of manuscripts, before Acknowledge

·

Basic reporting

The writing suffers somewhat in terms of clarity and succinctness. I’ve provided some guidance for possible editing in the manuscript.

It’s not clear to me why there is an apostrophe after the word seabirds in the title. Also, the title should perhaps be changed to something more descriptive of the work. A suggestion is:

Cross-species screening of microsatellite markers for use in individual identification in snow petrel Pagodroma nivea and Wilson’s storm petrel Oceanites oceanicus nesting in Antarctica.

The background and introductory material are sufficient to place the study within the marker development field, and literature references are relevant although there are inconsistencies in formatting that should be corrected.
Raw data are shared (thank you very much), although metadata are lacking and some locality information for each sample would be beneficial.
The structure of the paper is appropriate for the journal although I think the formatting for the major headings may be incorrect; the authors should carefully check the instructions to authors.
The figures and tables are relevant; I have provided some notes on the manuscript requesting clarification of certain points in the table. The figure is not aesthetically pleasing to me but I acknowledge that my art/design background gets in the way of my ability to fairly judge figure composition.

The submission is self-contained, including results relative to the focus of the paper, but there is some reference to population genetics parameters that appear to be assayable in a way that is outside the scope of the work described, which was to screen markers for individual ID in each species. The objectives of the paper should be followed closely, with reference to future broader uses placed within the discussion.

Experimental design

The research is within the Aims and Scope of PeerJ. The research is fairly well defined and knowledge gaps identified, but the paper would benefit from a specific statement of the objectives and not stray into a broader arena (see prior section). Original primary research within Aims and Scope of the journal.

I am unable to assess the appropriateness of the ethical treatment of the birds as I did not have access to the referenced letter (Letter no: NCAOR/ANT/ASPA/2014-15/01). Perhaps I simply didn’t find the correct document in the submission files.

The methods were sufficiently detailed that those in the laboratory I direct could follow them. I have provided edits on the manscript where clarity would benefit less experienced readers.

Validity of the findings

The findings appear to be valid as presented. However, justifying the findings with regard to marker resolution for individual ID could be strengthened by placing the results within the context of the guidelines recommended by Waits et al., or the use of PIDsib only (in the text) justified by some discussion.
I also recommend some discussion about the use of some of the markers (those > 200bp) for individual ID that may be applied to low quality sample types. It is well known that such markers are more subject to allelic dropout (see Sefc et al. 2003). One way to test this is to conduct sample reliability tests, such as the use of programs such as RELIOTYPE, and to screen paired high quality and low quality samples collected from the same individual.

Parameter metrics or at least P-values, even if not significant, should be reported (for example, a range of all locus by locus test values of linkage disequilibrium, or overall values or range, can be reported, along with p-values).

Regarding conclusions, the authors should carefully justify their statement that the statistical power for the markers used for the Wilson’s storm petrel is insufficient, whereas the snow petrel is. Guidance can be found in Waits et al., referenced in the manuscript.

---

## Round 0.2 · Minor Revisions

I am glad to acknowledge that most objections previously raised are now removed. There are however, a few points raised by Reviewer#1 that merit some attention, in particular the issue of the permit.
I also think that the discussion of the results, amplification failures, silent alleles and marker transferability is insufficient and probably some confusion remains, but maybe I am over demanding, asking for abeyance to forensic standards in this research field. Anyway, I dare to ask the consideration and distinction of two kinds of failure (which are particularly relevant to the question of di- versus tetranucleotidic STRs): on one hand there is the issue of DNA degradation/modification which is essentially a function of the amplicon size (and therefore explaining albeit only modestly the difference in the amplification of success rate of the two types) and the question of primer specificity vs (intra and interspecific) genetic diversity of primer-binding regions (which affects both).
I am sure that both criticisms can be successfully addressed and I will be gladly suggest acceptance of the revised version of your manuscript. In order to speed the process I am not resubmitting it to Reviewers again and I will make the editorial decision as soon as I receive it in the shortest possible time.

Reviewer 1 ·

Basic reporting

The authors have done much to improve the writing in the manuscript, in accordance with my previous comments as well as those of the other reviewers. There are however a few more points that should be addressed:

I noticed that much of the methods and results were primarily written in passive voice. The majority of journals currently request active voice, at least in the methods. While the author guidelines for this journal require ‘Established norms of academic writing within [our] field should be used’, there is not a specific mention of this particular point, so I will leave the validity of this comment up to the Editor.

Throughout the introduction, I noticed the use of the term "procellariiforms" to describe members of the order Procellariiformes, which seems appropriate when looking at other papers on this group, but here the lower-case form "procellariiformes" is also used, which seems inappropriate. Please revise to use the capitalized form when describing the order as a whole, and the first term I use in this comment when describing members of the order.

I have a few small comments on the text, which are detailed in the final section.

Experimental design

All details of experimental design seem appropriate, and are acceptable to me given the satisfaction of the other reviewers, who previously shared more detailed comments than my initial review.

Validity of the findings

As with experimental design, this section appears valid.

Additional comments

It appears that the permit information now provided by the authors consists of a denial, unless the sampling recommendation at the bottom of the document is sufficient to permit sampling without a separate approval for Broknes. Also, from what I can ascertain of Antarctica's geography, the two localities described in the permit letter are both found within the lower red box in Figure 1 - is there a separate permit for the area in the upper box? It seems like additional documentation is required.


Here are a few in-text edits to address. Line numbers refer to those in the clean revised PDF.

Line 64: Procellariiformes should be capitalized.

Lines 65-66: Change "species group" to something like "avian taxa"

Line 72: Add "the" after "Research on"

Line 74: Delete "few"

Line 210: "value was found" should be "values were found"

Line 225: For formality, "might not be robust enough" could be changed to "may not be sufficiently robust"

Lines 230-232: The sentence beginning with "Nevertheless" basically repeats lines 217-218 and can be deleted.

·

Basic reporting

The manuscript is well written and was conducted with two important Antarctica Seabirds, the results opens up perspectives for other studies. I consider that references are adequate, it show one figure with good quality and one table that authors included all suggestion of reviewers.

Experimental design

This study is original and brings important information about two flying species, Snow Petrel and Wilson's storm petrel, that there are poor knowledge about its genetic population. The markers standardization in this research can be used in others studies around Antarctica continent. The methods were well conducted and I believe that can replicate in other laboratories

Validity of the findings

The outcome of this study can be used by others scientific groups around Antarctica continent. The analyses were robust an appropriated, and the authors included several suggestion did by reviewers, improving the new version.
Thus, i have notthing to add.

---

## Round 0.3 · accepted · Accept

Although some issues of data quality still remain, since they are common in the field (namely not meeting forensic standards) and are not solvable at short notice, I think it can be published in the present form.

#